# Rotational Grazing Modifies *Rhipicephalus microplus* Infestation in Cattle in the Humid Tropics

**DOI:** 10.3390/ani13050915

**Published:** 2023-03-02

**Authors:** Gabriel Cruz-González, Juan Manuel Pinos-Rodríguez, Miguel Ángel Alonso-Díaz, Dora Romero-Salas, Jorge Genaro Vicente-Martínez, Agustin Fernández-Salas, Jesús Jarillo-Rodríguez, Epigmenio Castillo-Gallegos

**Affiliations:** 1Facultad de Medicina Veterinaria y Zootecnia, Universidad Veracruzana, Veracruz 91710, Mexico; 2Centro de Enseñanza, Investigación y Extensión en Ganadería Tropical, Facultad de Medicina Veterinaria y Zootecnia, Universidad Nacional Autónoma de México, Km. 5.5 Carretera Federal Tlapacoyan-Martínez de la Torre, Martínez de la Torre 93600, Mexico

**Keywords:** cattle, ectoparasites, control, grasslands, ticks

## Abstract

**Simple Summary:**

Ticks are one of the main problems in production units, mainly because they have become resistant to the chemicals used to control them. Several alternative methods to chemicals have been sought to control tick infestations in cattle, which are practical and friendly to the environment. In this work, we implement rotational grazing to combat ticks at the pasture level. We found that a 30-day rest period for pastures (without animals) is not enough to reduce the presence of ticks in animals but that a 45-day rest period does reduce the presence of ticks in cattle. These studies are critical since they would help cattle producers design better strategies that help reduce the use of chemical acaricides and the presence of chemicals in milk, meat, and the environment.

**Abstract:**

Rotational grazing has been mentioned as a potential tool to reduce losses caused by high tick loads. This study aimed: (1) to evaluate the effect of three grazing modalities (rotational grazing with 30- and 45-day pasture rest and continuous grazing) on *Rhipicephalus microplus* infestation in cattle, (2) to determine population dynamics of *R. microplus* in cattle under the three grazing modalities mentioned in the humid tropics. The experiment was carried out from April 2021 to March 2022 and consisted of 3 treatments of grazing with pastures of African Stargrass of 2 ha each. T1 was continuous grazing (CG00), and T2 and T3 were rotational grazing with 30 (RG30) and 45 d of recovery (RG45), respectively. Thirty calves of 8–12 months of age were distributed to each treatment (n = 10). Every 14 days, ticks larger than 4.5 mm were counted on the animals. Concomitantly, temperature (°C), relative humidity (RH), and rainfall (RNFL) were recorded. Animals in the RG45 group had the lowest count of *R. microplus* compared to the RG30 and CG00 groups; these results suggest that RG45 days of rest could be a potential tool to control *R. microplus* in cattle. Yet, we also observed the highest population of ticks on the animals under rotational grazing with a 30-day pasture rest. A low tick infestation characterized rotational grazing at 45 days of rest throughout the experiment. The association between the degree of tick infestation by *R. microplus* and the climatic variables was nil (*p* > 0.05).

## 1. Introduction

Ticks are one of the main threats to cattle production, affecting about 80% of livestock worldwide. These parasites generate losses ranging from 13.9 to 18.7 billion US dollars annually [1], mainly by affecting productive and welfare parameters. *Rhipicephalus microplus* (Canestrini, 1887) (Acari; Ixodidae) is the main ectoparasite affecting cattle in tropical, subtropical, and temperate areas of the world where it is a transmitter of pathogens such as *Babesia bovis*, *B. bigemina*, and *Anaplasma marginale* [2]. Worldwide control of this tick has mainly been based on therapeutic interventions using chemical treatments; however, these chemicals have developed acaricidal resistance in ticks and their ecological impact [3]. Non-conventional methods to control tick populations have been proposed to mitigate the effects of this resistance [4]. Among these are rotational grazing, in which the pasture manager maintains the grazing time cattle remains in the grazed section and determines the length of the recovery periods the animal stays off a pasture subdivision [5]. It aims to reduce the parasite–host interaction and has been mentioned as an ecological, profitable way and would also help optimize forage resources [6].

Rotational grazing as a means of tick control has received the attention of researchers for several years; nevertheless, [7] pointed out that most studies have been based on mathematical models, and there is little information on the effect of rotational grazing in the field. The preceding agrees with [8], who mentioned that population dynamics and dispersal of ticks in rotational grazing systems are complex and relatively unstudied. In field studies, it is observed that the effect of rotational grazing on ticks may depend on the recovery time of the paddock. Those rotational grazing with short periods of rest (20 days) showed higher tick infestations on-host than did those in continuous grazing. Some studies with longer paddock recovery times report that rotational grazing is a promising non-conventional strategy to control ticks.

On the other hand, generations of *R. microplus* are increasing throughout the year, a phenomenon linked with global warming [9], which justifies the need for current studies of the seasonal dynamics of this ectoparasite. A successful tick-control strategy will also depend on the interaction of biotic and abiotic factors that leads to seasonal population abundance that, in turn, determines tick behavior in each region. This knowledge plays a decisive role in an ectoparasite control program [9].

There is no information on the effect of rotational grazing with different resting times of paddocks on the control of *R. microplus* in cattle. The objectives of this study were: (1) to evaluate the effect of three grazing modalities (continuous grazing and rotational grazing with 30- and 45-day pasture rest) on *R. microplus* infestation in cattle, (2) to determine population dynamics of *R. microplus* in cattle under the three grazing modalities mentioned in the environmental conditions of the humid tropics.

## 2. Material and Methods

### 2.1. Study Site

The study took place in the Center for Teaching, Research and Extension in Tropical Livestock Production (CEIEGT) of the Faculty of Veterinary Medicine and Zootechnics of the National Autonomous University of Mexico (20°02′ N, 97°06′ W) [10], from April 2021 to March 2022.

### 2.2. Experimental Design

We tested three grazing management strategies: CG00, continuous grazing, where the animals roamed free in a single paddock without internal divisions; RG30, rotational grazing with 3-day and 30-day grazing and recovery times, respectively, with 11 paddocks of ≈0.18 ha each; and RG45, rotational grazing with 3-day and 45-day grazing and recovering times, respectively, with 16 paddocks of ≈0.12 ha each. The three experimental areas share the same latitude, and land irregularities are similar. Each treatment consisted of 2 ha of pastures where African star grass (*Cynodon nlemfuensis*) predominated and with natural tick infestations. Grazing by cattle has been the only use received by the pastures over the last 30 years. We used the flag technique to verify the presence of larvae in the three experimental areas before the start of the experiment. The number of larvae was very low and similar among pastures. Paddocks did not receive any anti-tick treatment before the investigation.

The experimental animals were thirty heifers between 8 and 12 months of age and had an average live weight of 182 ± 44 kg. We allocated ten heifers randomly to each treatment. Eight animals from each group were F1 (Holstein × Zebu) and two 5/8 × 3/8 (Zebu × Holstein). Another grouping criterion was coat color. The stocking rate for each experimental area at the beginning of the study was four animal units per hectare (au = 450 kg of live weight). Fifteen days before starting the experiment, all animals were treated against gastrointestinal parasites (albendazole), ticks, and flies (coumaphos) in order to start with similar tick loads. Report [11] indicates that the local tick populations resist amitraz, synthetic pyrethroids, chlorpyrifos, diazinon, and ivermectin. However, no study has evaluated tick susceptibility to coumaphos, a chemical that has been nil for at least 15 years.

### 2.3. Animal Management

Throughout the experiment, all animals received a feed supplement at a daily rate of 1 kg per head. We supplied water ad libitum to the animals. During the winter, the available grass decreased, so every other day, we supplemented the heifers with two bales of grass hay (*C. nlemfuensis* and *Brachiaria* sp., ≈22 kg each). There are no common areas between any of the three treatments. Each treatment had mobile and exclusive drinkers and feeders for the animals. The animals were not treated against ticks during the study; however, cattle were under medical supervision to monitor tick loads and clinical signs that might be present.

### 2.4. Tick Counts in Cattle

The count was carried out throughout the year every 14 days from 7:00 to 9:00 h. In total, we did 26 counts, beginning in April 2021 and ending in March 2022. With the aid of a compression ramp, we counted *R. microplus* only if its length was > 4.5 mm in each heifer. Ticks remained in place after counting. A qualified veterinarian examined the total body surface area of the right side and the number of ticks multiplied by two [12].

### 2.5. Climate Data Collection

Two weeks before and throughout the experiment, the environmental temperature (°C) and rainfall (mm) were recorded daily through the National Meteorological Service [13] database. The Weather Channel © mobile application provided the relative humidity data (RH, %) [14]. The climate is hot and humid with three climatic seasons: rainfall (June–September), winter (October–January), and dry (February–May). Our records (CEIEGT) and INEGI [10] indicated that the rainy season has temperatures ranging from 15 °C to 27 °C, rainfall is 715 mm, and relative humidity is 90–95%. The winter season (also known as “norths”) presents temperatures from 9 °C to 23 °C, with a total rainfall of 190 mm with a relative humidity of 30–90%. The dry season temperature varies from 11 °C to 29 °C, with rains of 150 mm and relative humidity of 20 to 80%.

### 2.6. Statistical Analysis

The data were analyzed using D’Agostino & Pearson, Anderson–Darling, Shapiro–Wilk, and Kolmogorov–Smirnov tests to determine normality and homogeneity of variances using StatGraphics 19.1.3 (StatPoint, Inc., Herndon, VA, USA). Tests showed that our data were not normally distributed. Tick counts for each treatment were compared using the Kruskal–Wallis test with Statistica 10.0 (StatSoft, Inc., Tulsa, OK. USA). A 95% confidence interval and a *p* value of less than 0.05 were considered. Environmental temperature, relative humidity, and rainfall were correlated with tick load using the Spearman test with Software R version 2021 (R Core Team, Vienna, Austria). The tick count in the bovines of each treatment was analyzed by descriptive analysis (Software R version 2021).

## 3. Results

Table 1 shows 26 counts of *R. microplus* teleogins (>4.5 mm). In the first three months (April, May, and June) after starting the experiment (six counts), the parasitic loads were very low and similar among treatments (*p* > 0.05) (Table 1). From count seven to count sixteen, corresponding to July to November, the animals in RG30 had the highest counts of *R. microplus* (*p* < 0.05) (Table 1). In the last ten samplings, corresponding to the November–March period, the tick count recorded in RG45 was significantly lower than RG30 and CG00 (*p* < 0.05).

The animals in the RG30 group had a higher cumulative parasite load at the end of the experiment with 13,352 teleogins, followed by animals from CG00 and RG45 treatments with 1882 and 660 teleogins. The dispersion patterns of parasitic loads among animals were different within each treatment. In total, 30% of the animals in RG30 and RG45 treatments concentrated 55% (7344/13,352) and 57% (1073/1882) of parasite loads, while in the CG00 treatment, the pattern was 42% (277/660). None of the animals showed health problems during the experiment.

The population dynamics of engorged ticks on each treatment showed variable patterns (Figure 1). Animals in the RG30 group presented the highest infestations of *R. microplus* ticks (>4.5 mm in length) throughout the year, and the population fluctuation showed five distribution peaks. The first peak of engorged females was in June and July, averaging 71.5 ticks per animal. The second and third peaks occurred during September and October, reaching an average of 188 ticks and 115 ticks per animal; fourth and fifth peaks occurred during January and February, with an average of 31 and 48 ticks per animal, respectively (Figure 1).

The experiment’s minimum and maximum environmental temperature fluctuated between 11.5 °C and 37.0 °C. Monthly relative humidity (RH) and precipitation were between 67 and 85% and 54.0 to 427.7 mm, respectively. There was no association between the degree of tick infestation by *R. microplus* and the climatic variables (*p* > 0.05).

## 4. Discussion

Rotational grazing has been reported as a viable alternative to control *R. microplus* in cattle. However, there is limited information on the effects of this non-chemical alternative at the farm level. Ours is the first report about the impact of three grazing management variations on *R. microplus* infestation in cattle. The evaluations that exist have been carried out in regions with different climates, different stocking rates and different types of pastures. However, these studies have reported significant findings that could help to understand the results obtained in the present report. In this study, we observed that reducing the length of the recovery period from continuous grazing or 45 days to a 30-day recovery stimulated the tick loads on heifers.

Animals in the RG30 group had the highest count of *R. microplus* compared to the RG45 and CG00 groups; indeed, the animals in treatment RG30 had a higher cumulative parasite load at the end of the experiment. Rotational grazing (with 20 days of rest) in *Cynodon dactylon* pastures was ineffective in reducing the parasitic loads of *R. microplus* on animals compared with continuous grazing [7]. The duration of the non-parasitic phase (in pastures) depends directly on climate and vegetation, which determine the abundance of the populations [15]. Under controlled field conditions, the average pre-hatching time is 42 days. Larvae show better activity to adhere to the potential host at 3 to 8 days post-hatching, indicating that the adequate time for the presence of viable and vigorous larvae in the pasture could be 45–50 days post detachment of engorged ticks [9]. If we consider in this study that the animals return to the paddock after 30 days, there would still be no viable larvae to infest. Still, in the next round, the animals would return after 60 days, when the larvae have 15–20 more days of their best viable age, which could be an adequate time for a high level of infestation under the conditions of this study. Further research is needed to determine the duration of the biological parameters of ticks under this grazing system and to consider other factors inherent to animal behavior. On the other hand, short-term rotational grazing induces a high stocking density because the number of paddocks increases, but the number of animals remains more or less constant. As the available pasture and area per animal decrease, the probability of the larva–host encounter would increase, leading to augmented tick loads on animals [7,15,16]. However, other factors can also influence infestations, such as animal behavior, cattle trampling, and the vegetation cover of pastures, among others.

Animals in the RG45 group, with a greater-density grazing system, had the lowest count of *R. microplus* compared to the RG30 and CG00 groups; indeed, the animals also had the lowest cumulative parasite load at the end of the experiment. These results suggest that RG45 days of rest could be a potential tool to control *R. microplus* in cattle. There is no information about the effect of rotational grazing with 45-day pasture rest on *R. microplus* infestation compared with grazing modalities such as 30-day pasture rest and continuous grazing. Unlike the RG30 group, the animals return to the paddock after 45 days, which could still mean a low percentage of viable ticks to infest. Still, in the next round, the animals would return after 90 days, when the larvae have 40–45 more days of their best viable age. Such length could be enough time for environmental conditions to damage the larvae, reducing the chances of infesting cattle.

The effect of rotational grazing on tick populations results from the impact that abiotic factors, type of pasture, and the recovery time can have on non-parasitic phases (pre-oviposition, oviposition, incubation, egg hatch, and larval maturation) of *R. microplus*. The vegetation architecture influences tick loads by protecting the larvae, as it climbs the pasture, from non-ideal environmental factors [9,15,17]; furthermore, it can also offer protection to other non-parasitic phases. In this study, and after each grazing period, the pasture in the RG45 could have been shorter and could expose larvae to harsh environmental conditions that increased the probability of dehydration. Although ticks have an essential capacity to resist prolonged starvation, the decrease in their energy reserves and overexposure to adverse climatic factors (such as high temperature and low humidity) [18] are among the leading causes of mortality in field conditions [19,20]. Some authors from different countries have repeatedly mentioned achieving completely tick-free pastures. For example, refs. [7,15,21] report 98, 105, and 136 to 192 days, respectively, because the larvae of *R. microplus* lose water and energy without having a source of nutrition. This variation highlights the need for studies under different conditions of vegetation cover (grazing times) and other times of the year to better understand the behavior of ticks under different pasture management systems and geographical conditions. In addition, since there is a higher stocking density than group RG30, the soil, pastures, and ticks are exposed to greater trampling, which could affect their survival. Further studies are suggested to determine these factors’ influence on ticks’ survival at the pasture level.

Remarkable data from the present study show that three out of ten animals in the RG30 and RG45 treatments maintained 55% and 57% of ticks, respectively. [16] reported a similar pattern, where 25 out of 36 animals were responsible for 50% of the total ticks. These results further reinforce the idea that animals have different susceptibilities or responses to tick infestations [22]. The above would help us to detect and treat only susceptible animals, as this would control around 55% of infestations and exert less selection pressure for resistance on ticks when treating animals. On the other hand, it is also essential to highlight behavior since some animals are leaders in the group and move ahead of others when the herd enters a new pasture; in this way, they can collect most of the tick larvae. Animal hierarchy can also explain why the tick load patterns are similar in the case of rotational grazing (55 and 57%) and lower (42%) for continuous grazing.

The growth and establishment of tick populations are directly related to the availability of hosts and the climate, such as temperature, humidity, and precipitation [23]. The observations during one year of the parasitic phase of *R. microplus* in bovines helped us to determine that the ectoparasite showed approximately five peaks in the RG30 and one in RG45 and group CG00. It is worth mentioning that these populations, being highly influenced by environmental conditions, can present various diapause times, which could manifest in the absence of marked peaks in some seasons and treatments. Previous studies in tropical and subtropical areas reported 3 to 4 peaks per year for this tick species in cattle [16,18,22]. However, a recent experiment showed the occurrence of five annual peaks of *R. microplus* in cattle [24], attributing temperature as a possible factor in the increase in peaks. In this regard, the cattle tick population dynamics from 40 years ago until now showed a tick population growth, with different peaks in a year depending on the seasonality (i.e., rainfall and dry seasons) or associated with the increase of the environmental temperature over the years [9].

The present study had no significant association between the parasite loads and the climatic variables as analyzed (*p* > 0.05). An aspect to highlight in the current experiment was the increase to a peak in group CG00 in winter, which is probably an effect of the cumulative number of larvae in the grasses that became adults in previous generations. This observation during the winter season in this experiment agrees with that reported by [16] in Brazil, where they found high peaks of *R. microplus* in continuous grazing, attributing to the quality of pasture and the nutritional status of cattle, inducing a lower probability of susceptibility of animals to ticks.

## 5. Conclusions

We observed the highest population of ticks on the animals under rotational grazing with a 30-day pasture rest. A low tick infestation characterized rotational grazing at 45 days of rest throughout the experiment. None of the climatic variables evaluated was related to tick loads in the experimental groups.

## Figures and Tables

**Figure 1 animals-13-00915-f001:**
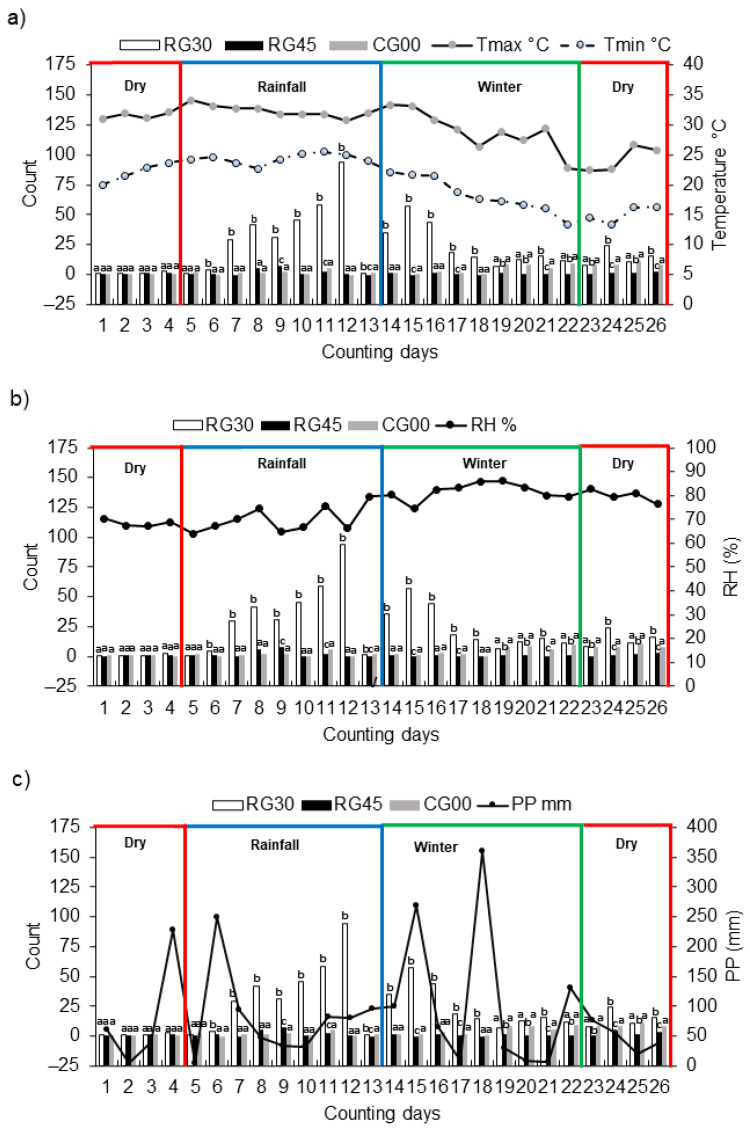
Number of *Rhipicephalus microplus* (mean ± standard deviation) under continuous grazing (CG00), rotational grazing with 30 and 45 days of rest (RG30 and RG45, respectively) on cattle (n = 10 per treatment), for 1 year (April 2021 to March 2022). Tmax = maximum temperature in °C; Tmin = minimum temperature in °C (**a**), °C = degrees centigrade, RH = relative humidity (%) (**b**), PP = pluvial precipitation (mm) (**c**). Within dates, bars with the same upper script letter are statistically equal at *p* < 0.05.

**Table 1 animals-13-00915-t001:** Number of *Rhipicephalus microplus* in cattle (>4.5 mm) under three grazing systems over a year.

Date	Sampling Number	CG00		RG30		RG45	
		Means ^1^	SD	Means	SD	Means	SD
02/04/2021	1	1.60 ^a^	1.58	1.40 ^a^	1.65	1.00 ^a^	1.41
17/04/2021	2	1.40 ^a^	1.65	1.40 ^a^	2.12	1.60 ^a^	2.07
01/05/2021	3	2.00 ^a^	2.49	2.20 ^a^	3.19	2.80 ^a^	3.68
18/05/2021	4	0.80 ^a^	1.69	6.20 ^a^	10.73	2.60 ^a^	4.01
01/06/2021	5	5.00 ^a^	4.74	2.20 ^a^	3.19	1.40 ^a^	2.50
15/06/2021	6	0.00 ^a^	0.00	8.60 ^b^	10.50	2.00 ^a^	5.66
29/06/2021	7	3.00 ^a^	3.16	59.20 ^b^	47.27	0.00 ^c^	0.00
13/07/2021	8	3.40 ^a^	3.53	83.80 ^b^	59.45	11.20 ^a^	14.58
27/07/2021	9	4.80 ^a^	3.16	61.80 ^b^	40.09	15.00 ^c^	7.62
11/08/2021	10	0.40 ^a^	0.84	90.80 ^b^	77.00	1.00 ^a^	1.94
24/08/2021	11	11.20 ^a^	8.12	117.20 ^b^	94.07	4.40 ^c^	6.24
08/09/2021	12	0.20 ^a^	0.63	188.40 ^b^	159.84	0.60 ^a^	1.35
20/09/2021	13	3.80 ^a^	5.29	162.40 ^b^	138.17	0.00 ^c^	0.00
05/10/2021	14	3.40 ^a^	3.41	70.80 ^b^	63.01	2.20 ^a^	2.90
18/10/2021	15	2.00 ^a^	3.27	115.00 ^b^	81.83	0.00 ^c^	0.00
03/11/2021	16	6.00 ^a^	6.18	88.00 ^b^	82.84	3.20 ^a^	4.54
16/11/2021	17	3.40 ^a^	3.66	36.60 ^b^	17.39	0.40 ^c^	0.84
30/11/2021	18	0.60 ^a^	1.90	29.00 ^b^	15.03	0.00 ^a^	0.00
14/12/2021	19	16.40 ^a^	13.16	13.00 ^a^	5.10	2.20 ^b^	2.57
28/12/2021	20	17.00 ^a^	5.52	25.00 ^a^	29.55	1.60 ^b^	2.27
11/01/2022	21	11.20 ^a^	7.19	31.00 ^b^	17.64	3.20 ^c^	3.16
25/01/2022	22	19.20 ^a^	14.37	23.00 ^a^	11.21	0.80 ^b^	1.93
08/02/2022	23	16.40 ^a^	11.46	16.20 ^a^	13.45	1.80 ^b^	2.57
22/02/2022	24	16.80 ^a^	9.53	48.00 ^b^	36.49	0.80 ^c^	1.40
09/03/2022	25	21.80 ^a^	5.37	22.20 ^a^	10.17	2.80 ^b^	3.43
23/03/2022	26	16.40 ^a^	8.04	31.80 ^b^	21.63	3.40 ^c^	2.32

^1^ Arithmetic mean of tick counts. ^a–c^ Values within a row with different superscripts differ significantly at *p* < 0.05.

## Data Availability

The database and the statistical analyses are available upon reasonable request.

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
