# Peer review of "Rotational Grazing Modifies Rhipicephalus microplus Infestation in Cattle in the Humid Tropics"

_animals, 2023, doi:10.3390/ani13050915_

Round 1
Reviewer 1 Report
The manuscript is interesting, and I agree we need to look for other alternatives to control ticks with environmentally friendly options.
The manuscript needs to be improved because comparing ticks counts and environmental variables is not always enough.
In my opinion, you must clarify the following:
Why do you consider three seasons i.e., Rainfall, winter, and dry? It is not clear the difference between rainfall and winter seasons.
Is the experimental area located in the same place? Are there any differences in altitude and land irregularities?
You need to give more detailed information about CG00 treatment; Is it a single paddock? Are animals free-roaming in 2 ha?
Were ticks counted and removed, or just counted? This information is critical because removing them may affect the number of ticks in the following generation.
Before the survey started, was it possible to observe any larvae in the paddocks? Or was there any treatment against ticks in the paddocks before the study? Or, was the paddock ticks-free because no animals were presented for months?
Have there been any information about productive parameters on heifers and the grass before, during, and after the study? Farmer prefers production to tick’s control.
Graphs should be presented with seasons and Temp, humidity, and precipitation trends during study periods.
Tick infestation on animals is influenced by environmental conditions and breed, color, and genetics inherited by animals. This information should be implemented in the manuscript to see the influence of other factors, especially skin and coat color.
Have you considered the diapause phenomenon in the variability of the tick numbers in each treatment or in general in each season?
Author Response
Reviewer 1
Comments and Suggestions for Authors
The manuscript is interesting, and I agree we need to look for other alternatives to control ticks with environmentally friendly options.
The manuscript needs to be improved because comparing ticks counts and environmental variables is not always enough.
Re: Thanks for the comments; we will improve the manuscript as suggested.
In my opinion, you must clarify the following:
Why do you consider three seasons, i.e., rainfall, winter, and dry? It is not clear the difference between rainfall and winter seasons.
Re: We consider the three main seasons that occur in the humid tropics of Mexico since each one presents different characteristics. We added a paragraph to clarify the difference between seasons (Lines 81 – 87).
Is the experimental area located in the same place? Are there any differences in altitude and land irregularities?
Re1: Yes.
Re2: There was no difference in altitude or land irregularities.
We add this description to the article (Lines 93 – 94)
You need to give more detailed information about CG00 treatment; Is it a single paddock? Are animals free-roaming in 2 ha?
Re: Yes, CG00 treatment was a single paddock, and animals roamed freely in 2 ha. We included this information in the current version of the manuscript (Lines 89 – 90).
Were ticks counted and removed, or just counted? This information is critical because removing them may affect the number of ticks in the following generation.
Re: The ticks were only counted. We now included this information in the reviewer manuscript version (Line 124).
Before the survey started, was it possible to observe any larvae in the paddocks? Or was there any treatment against ticks in the paddocks before the study? Or, was the paddock ticks-free because no animals grazed for months?
Re1: Yes, we observed larvae in the paddocks through the flag drag technique before the start of the experiment; however, the counts were low. This characteristic manifested itself during the first three months of starting the investigation (as we mentioned in the results).
Re2: There was no anti-tick treatment established or a period where the paddocks were without animals prior to the experiment.
We included this information in lines 97 -100.
Have there been any information about productive parameters on heifers and the grass before, during, and after the study? Farmer prefers production to tick’s control.
Re: Yes, we have the data of productive parameters on heifers (daily weight gain) during the year of the experiment. Average daily gains were: 0.305 g (RG30), 0.338 g (CG00), and 0.414 g (RG45). We did not include this information because we believe there is so much information to this article; the objectives are also different.
Graphs should be presented with seasons and Temp, humidity, and precipitation trends during study periods.
Re: Thanks. We modify the plots according to the recommendation and add them to the article (Line 172).
Tick infestation on animals is influenced by environmental conditions and breed, color, and genetics inherited by animals. This information should be implemented in the manuscript to see the influence of other factors, especially skin and coat color.
Re: We agree with the reviewer. Animals on each treatment were balanced by breed, color, and phenotype. We included this information in the manuscript in order to clarified this point (Line 104).
Have you considered the diapause phenomenon in the variability of the tick numbers in each treatment or in general in each season?
Re: Yes, this phenomenon is an essential biological part that occurs in the non-parasitic phase and is highly influenced by environmental conditions. Precisely, the evaluation of the population dynamics by seasons would allow us, among other things, to consider diapause as a vital part of the results obtained, especially in the winter season. We added a small paragraph in the discussion about this phenomenon (lines 256 – 258).
Reviewer 2 Report
Please see the attached report

Author Response
Reviewer 2
This article is of interest for integrated tick management and well written. It provides useful information concerning pastures management under tropical conditions. Field studies are not frequent, are time consuming, and give interesting data to enhance comprehension of tick biology and ecology. For these reasons, this article is worth publishing but some points need to be clarified.
Re: Thanks for the comments.
- General concept comments
Results of this fieldwork provide food for thought regarding pasture management but some initial information is required to fully analyze the data (monthly meteorological data, history of paddocks before cattle introduction for the study, information concerning potential tick resistance to acaricide...).
Re: We included this information as requested:
- Meteorological data has been added to graphs (Line 172).
- The use of paddocks before cattle introduction to the experimental groups has been mentioned in line 96 – 97.
- The status of acaricide resistance of the ticks in the study area was added in the line 107 - 110.
Discussion needs to be revised to be more accurate. Some results are compared with studies conducted in regions with different climates. Some parts are confusing or misinterpreted. However, I would be interested in continuing the discussion with the authors as the results raise interesting questions about the dynamics of tick populations in pastures.
Re: The discussion was reviewed and corrected to avoid confusion. A small paragraph was added to clarify the comparison between studies (Line 181 – 184).
Differences of resting times (30 days vs 45 days) may explain a part of the results but the height of the vegetation for example may also explain the observation. If scientific information is available, it would be interesting to provide data on the productivity of a Cynodon sp.. It would give information to see the impact of 10 heifers for 3 days on 0.12 ha, 0.18 ha or continuous grazing on grass height depending on the season. If the entire forage resource is consumed, the ticks will be exposed to direct sunlight which will decrease tick populations.
Re: Thanks for the note. Unfortunately, we don´t have the scientific information available for now on the data of the height of grass by seasons, however some collateral studies are being carried out to evaluate the influence of this characteristic and others such as trampling (due to the higher stocking rate), since it can also be an adverse factor that affected the pasture and ticks.
The wording was slightly modified since there is no information available for now; we changed “was very short” by “could have been shorter” (Line 226).
- Specific comments L76: it could be useful to add a table or a graph with monthly temperature, relative humidity and rainfall to compare local conditions to others papers.
Re: Information about climatic conditions during the study time was added to the tick count graph. The history of climatic parameters in the region have been added in the “Study site” section (Line 81 – 87).
L82: “annual” is missing before “rainfall”.
Re: The information was modified at the suggestion of the other reviewer to clarify the difference between seasons.
L84: it is important to add information concerning paddocks history before the beginning of the study. Were they all grazed by cattle before the beginning or were they kept out of animals – and for how long? It is of interest to know if, all or some, paddocks were infested by tick larvae when animals were introduced as it will affect the infestation of animals during the trial.
Re: Information was added as requested (Line 96 – 97).
It’s also important to know if there are common areas for the batches RG30 and RG45 (trough, feeding area...). For example, are there a trough in each paddock or is there a common drinking trough for all paddocks in lot RG30 (and for RG45)? If there is a common trough, do animals use corridor to go drinking? If yes, are these corridors grassy – that can represent a source of tick infestation? It’s important to know these details as they can affect tick population dynamic.
Re: There were no common areas among treatments (no drinkers, feeders or corridors). Information was added as requested (Line 115 – 118).
Concerning acaricidal treatment, could authors provide information on resistance to Coumaphos for the local tick population as it is described in Mexico.
In addition, it’s not mentioned that animals were treated during the trial despite high infestations. Could authors confirm this point?
Re1: Information was provided as requested (Line 107 – 110).
Re2: The animals were not treated against ticks during the study. Information was added in lines 117 -118.
L147 (and L224): it’s not correct to write that the 5 peaks of tick correspond to the 5 generations of tick per year. These 5 peaks are grouped within 8 months that is very short for 5 generations. Moreover, delays between the second and the third peak is 40 days and between the 4th and the 5th peak is 42 days. If we count 22 days for the parasitic stage, it makes 18 and 20 days for the prehatching periods that is not described in the literature.
Re: We agree with the reviewer. We deleted the paragraph regarding “generations” (Line 163) so as not to create any confusion. A study is being carried out to evaluate the biology of ticks and the duration of each non-parasitic phase according to each treatment.
L174-177: not clear, need to be rewritten. I cannot find this information in the cited article
Re: We rewrote the paragraph to give more straightforward and more appropriate information (Lines 193 – 203 and Lines 215 – 219)
L178-183: not clear. Stocking density is 22.2 a.u / 0.18 ha for the RG30 and higher (33.3 a.u. / 0.12 ha) for RG45. This would suggest that larva-host would increase for RG45. But it’s the opposite in the results and in the following paragraph.
Re: We agree with the reviewer; our results were different from what is mentioned in the reference. In this regard, we consider that other factors could have influenced the decrease in infestations in RG45. We add a small paragraph on lines 207 - 208 to clarify this point.
L196-198: this is an interesting point and maybe a part of the explanation for the results. It would be of interest to give the information concerning pasture height for CG00 and after grazing period for RG30.
Re: Unfortunately, we don´t have the scientific information available for now on the data of the height of grass by season, however, some collateral studies are being carried out to evaluate the influence of this characteristic and others, such as trampling (due to the higher stocking rate), since it can also be an adverse factor that affected the pasture and ticks.
The wording was slightly modified since there is no information available for now; we changed “was very short” by “could have been shorter” (Line 226).
Also, we consider that the cattle trampling in the 45-day rest group (due to the higher stocking density) may also be an adverse factor that affected the pasture and ticks. We added a small paragraph about it (Line 237 – 240).
L203 : it’s tricky to compare results with studies done in north Argentina (Nava 2013 and Mastropaolo 2017) and climate conditions – and tick population dynamic – are very different. Unless I’m wrong, Nicaretta concluded that 105 days are required to reduce tick populations in a pasture, not 60 days.
Re: We agree with the reviewer. We have modified the paragraph to avoid confusion (Line 231 – 234).
L205-209 : observations of Desquesnes may explain results during the summer but hatching period is longer in winter and can not explain results observed during the rest of the trial.
Re: We agree with the reviewer. We removed the paragraph where we mentioned Desquesnes.
L209-211 : Cruz (2020) conducted the trial in paddock of Urochloa decumbens not Cynodon nlemfuensis. It seems that, in his study, hatching time varied from 30 days to 63 days. 26 days was in controlled conditions (BOD chamber). Whatever, 13 to 19 days is very short to reduce larval populations and explains the observed results.
Re: We agree with the reviewer. We eliminated the paragraph to avoid confusion.
L217-218: genetic variations may explain the differences in tick infestations between animals but behavior should also be considered. Some animals are leader in the group and move ahead of other animals when the herd enters a new paddock. By this way, they can collect the majority of tick larvae. This may explain why patterns of tick loads are similar in case of rotational grazing and different (and lower) for continuous grazing.
Re: We included this information as requested (Line 247 – 251).
L224-226: it’s not correct to write that there are 5 generations of ticks in the RG30 (see L147) and one in RG45 and CG00. One generation per year would suggest that non parasitic period would have to last one year (minus 22 days). It is more likely that the number of tick generations is the same for all groups – as it depends on meteorological conditions – but the peaks are smaller and therefore go unnoticed for RG45 and CG00.
Re: We agree with the reviewer and changed “generations” to “peaks (Line 255).
L236: “(P > 0.05)” not “(P < 0.05)”
Re: Changed as suggested (Line 267).
Round 2
Reviewer 2 Report
That's fine for me with the revised version. Just a last question to be sure. L115-117, "there are no common areas between any of the three treatments. Each treatment had mobile and exclusive drinkers and feeders". I understand that drinkers are moved with the animals for RG30 (and RG45) and there is not a common drinker or feeder area for all RG30 paddocks. Am I right ?
Thanks
Author Response
Reviewer 2
That's fine for me with the revised version.
Re: Thanks.
Just a last question to be sure. L115-117, "there are no common areas between any of the three treatments. Each treatment had mobile and exclusive drinkers and feeders". I understand that drinkers are moved with the animals for RG30 (and RG45) and there is not a common drinker or feeder area for all RG30 paddocks. Am I right ?
Re: Yes. It is right.